# *GuoFeng*: A Discourse-aware Evaluation Benchmark for Language Understanding, Translation and Generation

## Abstract

Modeling discourse – the linguistic phenomena that go beyond individual sentences, is a fundamental and challenging problem in natural language processing (NLP). However, existing evaluation benchmarks mainly focus on the evaluation of inter-sentence properties and overlook important discourse phenomena that cross sentences. To bridge the gap, we propose a GuoFeng benchmark that can evaluate intra-sentence discourse properties across a diverse set of NLP tasks, covering understanding, translation, and generation. GuoFeng consists of 9 document-level testsets in the literature domain, which contain rich discourse phenomena (e.g. cohesion and coherence) in Chinese and/or English. For linguistic analysis, we also propose a diagnostic test suite that can examine whether the target models learn discourse knowledge. We evaluate 17 general- and in-domain models based on Transformer and advanced pretraining architectures, showing that fine-grained pretraining based on document-level training data consistently improves the modeling of discourse information. We will release the datasets, pretrained models, and leaderboard, which we hope can significantly facilitate research in this field.

## 1 Introduction

To evaluate the general performance of models, previous work proposed a variety of benchmarks, covering different tasks and languages such as GLUE (Wang et al., 2018), CLUE (Xu et al., 2020) and XGLUE (Liang et al., 2020). However, existing benchmarks pay little attention to discourse phenomena, which is a fundamental and challenging problem in natural language processing (NLP) (Kevitt et al., 1992). The natural language generally consists of meaningful, unified, and purposive groups of sentences, which are organized as a whole according to discourse properties (Cook, 1989). As shown in Figure 1, the discourse property manifests in two ways: (1)

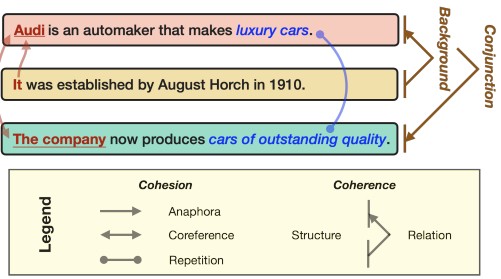

Figure 1: Discourse definition and example. The concept of discourse is detailed in Appendix §A.

**cohesion**, where the dependency between words or phrases makes them logically and consistently connected; (2) **coherence**, where the structural relation between segments or sentences enables them semantically and meaningfully composed.

To bridge the gap, we introduce a GuoFeng benchmark for the target evaluation on the discourse modeling. GuoFeng comprises three parts:

- **GuoFeng Benchmark**: It consists of nine Chinese/English discourse-aware tasks covering a broad range of NLP tasks (understanding, translation, and generation), data quantities (from 26.4K to 2.4M), and difficulties. Besides, most benchmarking datasets are newly created in this work.

- **GuoFeng Diagnostic Dataset**: To understand the discourse information learned by models, GuoFeng also includes a dataset of hand-crafted 600 examples for probing trained models. Each instance in the dataset is a contrastive pair, where the correct candidate is the original instance in

| Task | Description | Metric | Dataset | | | Language |
|---|---|---|---|---|---|---|
| | | | **# Train** | **# Test** | **Domain** | |
| | | | *Understanding Task* | | | |
| **SI** | **S**peaker **I**dentification | F1 | 48.0K | 17.5K | novel | zh |
| **ZPR** | **Z**ero **P**ronoun **R**ecovery | F1 | 2.2M | 8.1K | mixed | zh |
| **MRC** | **M**achine **R**eading **C**omprehension | Accuracy | 26.4K | 6.5K | composition | mzh, czh |
| | | | *Translation Task* | | | |
| **NT** | **N**ovel **T**ranslation | BLEU | 1.9M | 1.3K | novel | zh→en |
| **CCT** | **C**lassical **C**hinese **T**ranslation | BLEU | 778.1K | 5.3K | dianji | czh→mzh |
| **PT** | **P**oetry **T**ranslation | BLEU | 47.1K | 2.7K | poetry | zh→en |
| | | | *Generation Task* | | | |
| **TE** | **T**ext **E**xpansion | BLEU | 2.4M | 10K | book | en |
| **TI** | **T**ext **I**nfilling | PPL | 233K | 10K | book | zh |
| **TC** | **T**ext **C**ompletion | PPL | 233K | 10K | book | zh |

Table 1: An overview of our discourse-aware evaluation benchmark, covering language understanding, translation and generation. All datasets consist of document-level texts in the literature domain, which are rich in discourse phenomena. Eight of them are newly created by us and one is expanded based on existing corpus (i.e. MRC). It covers three languages: English (en), Modern Chinese (mzh/zh) and Classical Chinese (czh). We only report commonly-used evaluation metrics here while more metrics are discussed in Appendix §B. "#" means the number of instances (e.g. sentences, pairs or documents). "Test" represents both validation and testing sets.

the benchmark and the incorrect one is a perturbation by modifying discourse devises or structures in the correct candidates.

- **GuoFeng Training Data**: We release a large-scale, document-level data (400G) in Chinese and English, which is in the same literature domain with the benchmark. The training data enables fine-grained pretraining to better model discourse information required by the benchmark.

To better understand challenges posed by GuoFeng, we conduct experiments on a variety of state-of-the-art models, including Transformer and pretrained models. We found that these tasks display different levels of difficulty, resulting in different behaviors and performances across models. Furthermore, the fine-grained pretraining based on the document-level and discourse-rich GuoFeng data improves performances particularly on cohesive translation and coherent generation. However, the best models still achieve a fairly low absolute score, highlighting the difficulty of modeling discourse.

There are three **main contributions** in this work:

- **Challenging Tasks**: We propose a diverse set of discourse-aware tasks to evaluate monolingual and cross-lingual models' ability to understand and generate nature language.

- **Considerable Resources**: We build and release a variety of discourse-aware resources, including benchmarking datasets, diagnostic test suite, large-scale pretraining corpus and discourse-aware pretrained models.

- **Comprehensive Comparisons**: We systematically compare many advanced pretraining methods on the benchmark, and identify current challenges in discourse modelling for future exploration.

## 2 DISCOURSE-AWARE TASKS

To comprehensively evaluate the target models, GuoFeng covers three types of NLP tasks, including language understanding, translation and generation. We design the benchmarking tasks using the following criteria: (1) our tasks should measure the ability of models to handle discourse phenomena, thus we define discourse-related tasks at different levels of difficulty; (2) our datasets should contain rich discourse phenomena, thus we build document-level datasets with whole contexts extracted from literary texts. To this end, we introduce nine discourse-aware tasks, which are representative of challenging NLP tasks, and easily applicable to real-world situations.

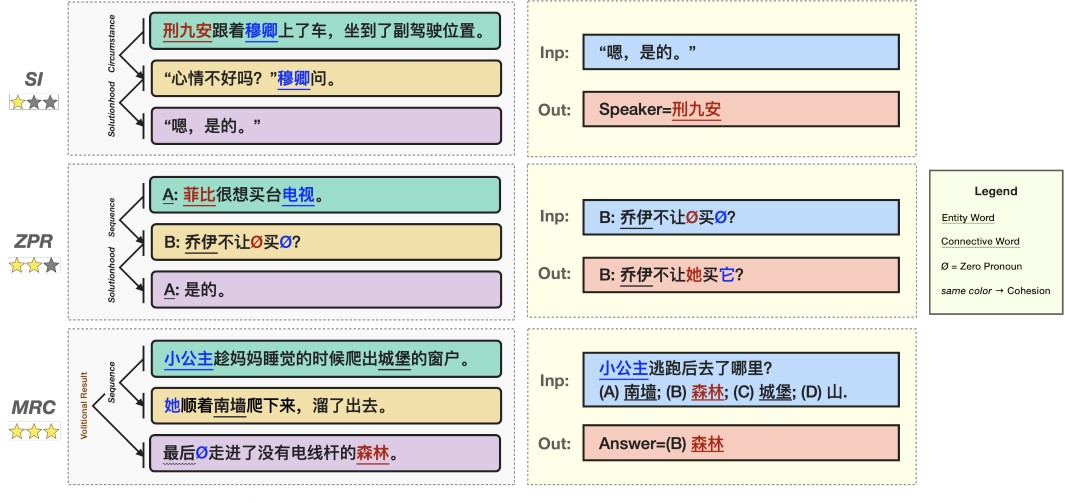

Figure 2: Illustration of the proposed **understating tasks** in terms discourse properties and task definition. As seen, SI needs to recognize named entity and resolve coreference. While ZPR demands the further ability to tackle zero anaphora and gender identification. MRC is the hardest because it should fully understand coherence (e.g. discourse structure based on temporal relation) apart from cohesion in previous tasks. English translations of example sentences are listed in Table 6.

Accordingly, the benchmark contains a collection of nine datasets in Chinese and/or English: eight of which are newly created, and one is expanded based on existing data. Table 1 lists the details of the benchmark, where each task contains training, validation, and testing sets. In the following subsections, we mainly introduce task definition, discourse awareness, and data construction. More details such as additional evaluation results are described in Appendix §B.

## 2.1 LANGUAGE UNDERSTANDING TASKS

NLU aims to analyze what human language means, containing various tasks such as natural language inference and story comprehension. Discourse is one of the fundamental problems for NLU models. It is difficult to determine the referents of pronouns and definite noun phrases, and understand elliptical sentence fragments, as well as a host of other long-range language phenomena that have not even been adequately characterized much less conquered (Bates, 1995). As shown in Figure 2, we classify tasks into three difficulty levels according to the length of contexts and the amount of knowledge, required for discourse modeling.

**SI (Speaker Identification)**    Given a paragraph that may contain an utterance and the surrounding context, SI aims to identify the corresponding speaker(s) for the utterance or the content within quotation marks if no speaker exists. To archive this goal, models need to examine the existence of quotes, recognize named entities or phrases that can serve as speakers, and resolve coreference. We construct the dataset that contains 66K instances based on eighteen Chinese novels. Unlike previous SI datasets such as P&P (He et al., 2013) in which all speakers are entities, speakers in our dataset can also be phrases, pronouns, or multi-entities.

**ZPR (Zero Pronoun Recovery)**    ZPR aims to recover omitted pronouns in terms of position and form, according to its anaphora information in the given sentence (Yang & Xue, 2010; Zhang et al., 2019b; Song et al., 2020). The BaiduKnows is a widely-used Chinese ZPR corpus, which contains only 5K human-annotated sentences extracted from a Q&A forum (Zhang et al., 2019b). The insufficient data limits the investigation of model performance on ZPR. Inspired by Wang et al. (2016), we automatically built a large-scale training set from Chinese-English movie subtitles using word alignments. For a clean test set, we hire experts to manually annotate 8K sentences covering five domains (i.e. 1.7K novel, 2.2K movie subtitle, 1.2K Q&A forum, 1.6K news, and 1.5K resume). The label set contains 30 Chinese pronouns according to person, number, and gender.

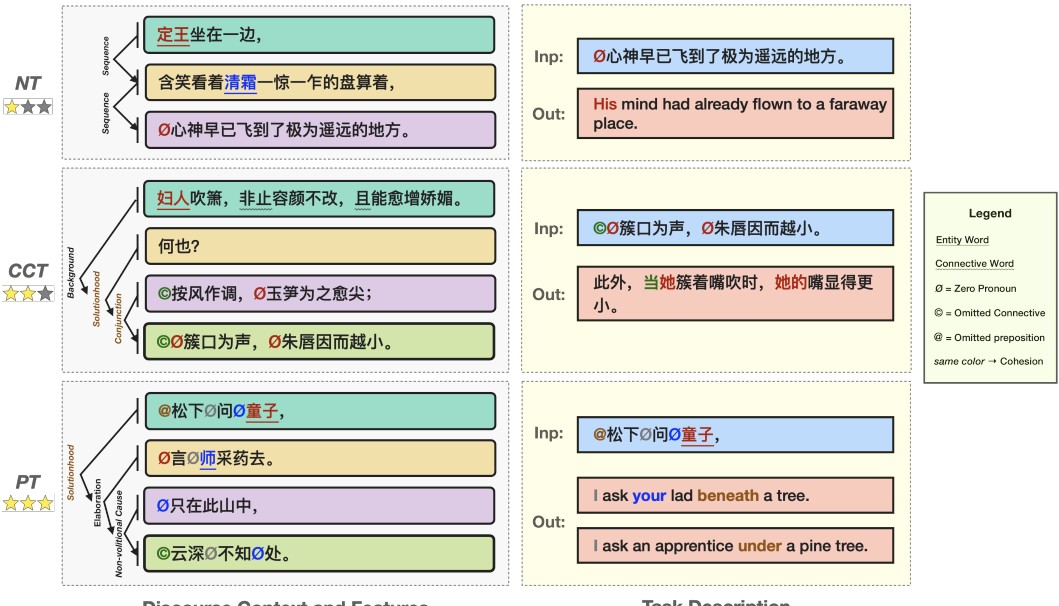

Figure 3: The illustration of the proposed **translation tasks** in terms of discourse properties and task definition. As seen, a variety of elements may be omitted in the Chinese input but should be recalled in English translation. NT mainly deals with zero pronouns while CCT needs to further tackle omitted connective words that are the marker of discourse structure. PT is the most difficult task because even prepositions could be further omitted. English translations of example sentences are listed in Table 6.

**MRC (Machine Reading Comprehension)**    The goal of MRC is to answer questions based on the understanding of its meaning given an unstructured text (Liu et al., 2019a; Zeng et al., 2020). We collected the Haihua2021 corpus, which contains 8K articles extracted from reading comprehension tests in primary/high school examinations.[1] Each article is followed by at least one question with 2~5 choices and one correct answer. We manually create 2K articles as an additional supplement. Different from previous benchmarks based on Wikipedia texts (Cui et al., 2019) or Chinese idioms (Zheng et al., 2019), the Haihua2021 corpus is in the literary domain (i.e. modern/ancient composition and poetry) that contains rich discourse phenomena. Different from the $C^3$ benchmark (Sun et al., 2020) where problems are collected from Chinese-as-a-second-language examinations, this dataset is extracted from more challenging examinations designed for native speakers.

## 2.2 LANGUAGE TRANSLATION TASKS

Language translation is a sequence-to-sequence generation task to translate text from one language to another. Discourse information is important for document-level translation to produce cohesive and coherent translations (Wang et al., 2017; Bawden et al., 2018). As shown in Figure 3, we design three translation tasks of increasing hardness, which differ in the conciseness of source sentences in Chinese. The more concise the Chinese text, the more discourse information is needed for translation.

**NT (Novel Translation)**    The significant challenges for translating novels are entity consistency, anaphora resolution, and lexical choice (Matusov, 2019). We build a document-level Chinese-English corpus, which is extracted from web fictions (150 books in 14 genres). We manually align them at both document and sentence levels. Different from previous document-level MT datasets such as LDC[2] and OpenSubtitle[3] from the news and movie subtitle domains, ours is the first literature-domain MT corpus containing richer linguistic phenomena especially in discourse.

---

[1]https://www.biendata.xyz/competition/haihua_2021.
[2]https://www.ldc.upenn.edu.
[3]https://opus.nlpl.eu/OpenSubtitles-v2018.php.

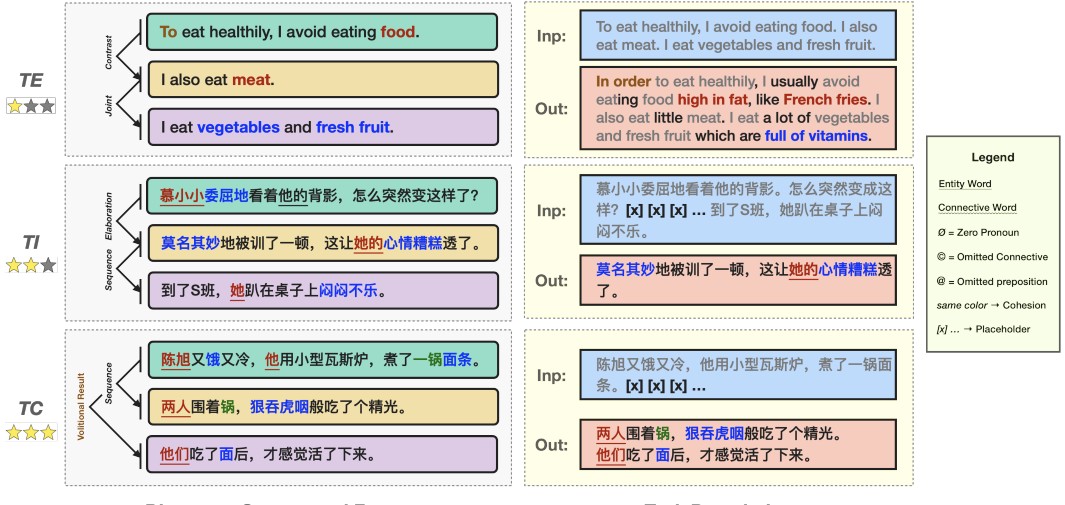

Figure 4: The illustration of the proposed **generation tasks** in terms of discourse properties and task definition. As seen, discourse structure and main contents have been specified in TE, thus the task needs to generate cohesive words. While TI should further consider cohesion relations when generating a whole sentence based on the previous and following ones. TC is the most difficult because it needs to generate more sentences with a unified structure. English translations of example sentences are listed in Table 6.

**CCT (Classical Chinese Translation)** Compared with modern Chinese as in novel translation, classical Chinese texts contain more implicit connectives and pronouns, which require discourse information for information recovery. We construct a document-level classical-modern Chinese dataset, extracted from Chinese classics across history branch.[4] Different from the NiuTrans Classical-Modern corpus[5] that has no discourse context, ours maintain the original context.

**PT (Poetry Translation)** Poetry translation is regarded as one of the hardest tasks in computational linguistics, or even artificial intelligence in general (Genzel et al., 2010; Ghazvininejad et al., 2018). Chinese poetry is more concise with implicit coherence, which is generally reflected through situational context and contextual context. For example, Chinese poetry does not use any cohesive means, but the semantic is still clear. We build a document-level Chinese Poetry-English corpus, covering different types of Chinese poetry (e.g. Shi, Ci, Qu, and Fu) translated by famous translators.

## 2.3 LANGUAGE GENERATION TASKS

Language generation is a sequence generation task to produce text based on a given context (Reiter & Dale, 1997). Generating long and coherent text is an important but challenging task, particularly on lexical cohesion (Wanner, 1996; Guan et al., 2021). As shown in Figure 4, we design three representative generation tasks that differ in degrees of freedom. The more open-ended the generation task, the more difficult to generate accurate cohesive devices and discourse structure.

**TE (Text Expansion)** We also define a new task, which has been seldom studied previously: given a predefined text, the goal of TE is to insert appropriate words, phrases, or clauses for adding more details and deepening the meaning, while retaining coherence and cohesiveness. We use a semi-automatic generation method to obtain large-scale training data. The raw data are extracted from English books detailed in Table 14 Specifically, we use the Stanford Parser[6] to produce the syntactic tree of a text, and then manually design some rules to delete the modifier words and phrases in the text. We use the remaining words as the input and predict the dropped modifier. Since some

---

[4]https://en.wikipedia.org/wiki/Chinese_classics.
[5]https://github.com/NiuTrans/Classical-Modern.
[6]https://github.com/stanfordnlp/CoreNLP.

| Type | Input/Source/Previous | Hypothesis/Target/Next |
|---|---|---|
| | *Understanding Task* | |
| **Cohesion** | **Correct**: 菲比很想买台电视。
(*Phoebe really wants to buy a TV.*)
**Incorrect**: 吉姆想买台电视。
(*Jim really wants to buy a TV.*) | 但乔伊不让她买它。
(*But Joey won't let her buy it.*) |
| **Coherence** | **Correct**: 小公主爬出城堡。最后她躲进了森林。
(*The little princess escaped from the castle. In the end she hid in the forest.*)
**Incorrect**: 小公主爬出城堡。然而她没躲进森林。
(*The little princess escaped from the castle. However, she did not hide in the forest.*) | 小公主逃跑后去了哪里？
(*Where did the little princess go after she escaped?*)
(B) 森林
(B) (*Forest*) |
| | *Translation Task* | |
| **Cohesion** | **Context**: 定王含笑看着清霜。
(*King Ding looked at Qingshuang with a smile.*)
**Current**: 觉得清霜很滑稽。 | **Correct**: He thinks Qingshuang is funny.

**Incorrect**: I think the clear cream is funny. |
| **Coherence** | **Context**: 丽莎没上过学。
(*Lisa did not go to school.*)
**Current**: 她只能写到这个水平。 | **Correct**: Thus, she can only write to this level.

**Incorrect**: She can only write to this level. |
| | *Generation Task* | |
| **Cohesion** | 慕小小被莫名其妙训了一顿。
(*Mu Xiaoxiao was reprimanded for no reason.*) | **Correct**: 她趴在桌子上闷闷不乐。
(*She was sullen on the table.*)
**Incorrect**: 她趴在桌子上开心坏了。
(*She was overjoyed lying on the table.*) |
| **Coherence** | 天降大雨，陈旭又饿又冷。
(*It rained heavily, and Chen Xu was hungry and cold.*) | **Correct**: 于是他赶紧煮了一锅面。
(*So he quickly cooked a pot of noodles.*)
**Incorrect**: 然而他赶紧煮了一锅面。
(*However, he quickly cooked a pot of noodles.*) |

Table 2: The illustration of the proposed test suite. We design each contrastive instance with correct and incorrect discourse markers in terms of cohesion and coherence. Tested systems are asked to rank candidates according to their model scores.

delete operations may produce ill-formed text, we filter out the training instances if the remaining text has a large perplexity measured by a language model.

**TI (Text Infilling)** The TI task aims to predict a text snippet given its surrounding context (Zhu et al., 2019). To evaluate the discourse-level model capability, we focus on the sentence infilling task that predicts a missing bridge sentence $x_0$ given two preceding sentences ($x_{-2}$ and $x_{-1}$) and two subsequent sentences ($x_1$ and $x_2$) (Huang et al., 2020; Cai et al., 2020). We build a new TI dataset by extracting consecutive 5-sentence paragraphs from Chinese web fictions used in the NT task.

**TC (Text Completion)** The TC task is to predict a writing continuation given a preceding prompt. We focus on multi-sentence paragraph completion for a target evaluation of discourse modeling, which completes a multi-sentence paragraph $x_{s:e}$ given its leading sentence $x_s$. We use the same data collected for the TI task to construct the TC dataset. Specifically, given a sentence $x_{-2}$, we aim to predict the concatenation of $x_1$, $x_0$, $x_1$, and $x_2$.

## 3 DISCOURSE-AWARE TEST SUITE

The general-purpose automatic metrics (e.g. BLEU and PPL) may be not sufficient to distinguish model performance in terms of discourse (Wong & Kit, 2012; Müller et al., 2018; Voita et al., 2018; 2019; Lin et al., 2011). To better measure the ability of models on discourse modeling, we handcraft a discourse-aware test suite that is complementary to general evaluation. More specifically, each instance is a contrastive pair with correct and incorrect propoteties in terms of cohesion or coherence. We directly employ the original texts in the benchmark testset as correct candidates. For incorrect candidates, we add noises to the correct one by modifying its discourse devices (cohesion) and structures (coherence). Given the input and output, we can calculate the model score candidates in each instance. Accordingly, we assess models on their ability to rank the correct candidate higher than the incorrect one. To end, we totally construct 600 instances, and each task/type contains 100.

| # | Model | Language | Size | Task | Corpus | |
|---|-------|----------|------|------|--------|--|
| | | | | | **Size** | **Sources** |
| 1 | BERT (base) | zh | 110M | U, T | 1.5GB | Wiki |
| 2 | RoBERTa (base) | zh | 110M | U, T | 15GB | Wiki, EXT Corpus |
| 3 | RoBERTa (large) | zh | 340M | U, T | 15GB | Wiki, EXT Corpus |
| 4 | AnchiBERT (base) | zh | 102M | U, T | 1.5GB | Classical Chinese |
| 5 | MengziBERT (base) | zh | 103M | U, T | 300GB | Wiki, Common Crawl |
| 6 | BART (large) | zh, en | 406M | U, T, G | 200GB | Wiki, WuDao Corpus |
| 7 | mBART (CC25) | zh, en, etc. | 610M | T | 1.4TB | Common Crawl |
| 8 | GPT2 (base) | zh | 102M | G | 14GB | CLEU Corpus |
| 9 | GPT2 (large) | en | 762M | G | 40GB | Web Text |
| 10 | T5 (base) | zh | 231M | G | 14GB | CLEU Corpus |
| 11 | T5 (large) | en | 770M | G | 745GB | C4 |
| 12 | GuoFeng (family) | zh, en | – | U, T, G | 400GB | Literature |

Table 3: Summary of pretrained models used as baseline, varying in model architecture, parameter scale, training data, and targeted task (i.e. understanding, translation, and generation). #1∼11 are publicly available. #12 denote a series of pretrained models that are continuously trained on our literature-domain data initialized by corresponding parameters in #1∼11.

Considering three types of tasks, we employed different strategies, varying in the place that is modified. Table 2 illustrates how we design contrastive pairs:

- **Understanding Tasks**: for incorrect candidate, we add noises to the input $x$ (i.e. $x \rightarrow x'$) while keeping hypothesis $y$ unchanged. Thus, each instance contains correct $(x, y)$ and incorrect $(x', y)$ candidates. We calculate model scores by feeding them into corresponding models.

- **Translation Tasks**: inspired by Bawden et al. (2018) and Voita et al. (2019), we add noises to the target translation to generate an incorrect candidate (i.e. $y \rightarrow y'$) while keeping the source input $x$ unchanged. We ask model to score two candidates $(x, y)$ and $(x, y')$ via force-decoding method.

- **Generation Tasks**: for TE task, we add noises to expanded sentences (i.e. $y \rightarrow y'$) in a text while keep the predefined one $x$ unchanged. Given $(x, y)$ or $(x, y')$, models can calculate their models scores. In TI and TC, we regard generation models as the language model, which can directly calculate perplexity of a sequence of sentences. Thus, we add noises to any sentences in a text.

## 4 BENCHMARK RESULTS

### 4.1 BASELINES

**Plain Models** We use the Transformer (Vaswani et al., 2017) with *base* and *big* configurations as our plain models. We use the Adam optimizer with $\beta_1 = 0.9$ and $\beta_2 = 0.98$, and employed large batching Ott et al. (2018) for model training. We set the max learning rate to 0.0007 and warmup-steps to 16000. All the dropout probabilities are set to 0.3.

**Existing Pretrained Models** We systematically compare SOTA pretraining models on our constructed discourse-aware benchmark, including BERT (Devlin et al., 2019), RoBERTa (Cui et al., 2020), AnchiBERT (Tian et al., 2021), MengziBERT (Zhang et al., 2021), BART (Lewis et al., 2020; Shao et al., 2021), mBART (Liu et al., 2020), GPT2 (Radford et al., 2019; Zhao et al., 2019), and T5 (Raffel et al., 2020; Zhao et al., 2019). Table 3 shows the summary information of the pretrained models detailed in Appendix §C. We fine-tuned these public models on the corresponding datasets for downstream tasks For translation tasks, we use BERT-based pretrained models (e.g. BERT, RoBERTa) to initialize the encoder of NMT models. We choose the hyper-parameters based on the performance on the validation set for each model. We fine-tune each model twice and report the averaged test results. The fine-tuning hyper-parameters are detailed in Table 13.

**Discourse-Aware GuoFeng Pretrained Models** The frequencies and types of discourse phenomena vary in different domains (Yang et al., 2015), leading to differences in model behavior and quality

| Model | Understanding | | | Translation | | | Generation | | |
|---|---|---|---|---|---|---|---|---|---|
| | SI$\uparrow$ | ZPR$\uparrow$ | MRC$\uparrow$ | NT$\uparrow$ | CCT$\uparrow$ | PT$\uparrow$ | TE$\uparrow$ | TI$\downarrow$ | TC$\downarrow$ |
| *Plain Models* | | | | | | | | | |
| Transformer (base) | 9.1 | 10.8 | 39.9 | 22.0 | 43.9 | 6.6 | 20.2 | 15.5 | 20.0 |
| Transformer (big) | 4.4 | 11.1 | 38.4 | 22.5 | 44.8 | 6.6 | 22.4 | 14.7 | 18.5 |
| *Existing Pretrained Models* | | | | | | | | | |
| BERT (base) | 85.1 | 17.4 | 49.2 | 21.6 | 57.0 | 6.7 | - | - | - |
| AnchiBERT (base) | 81.3 | 23.2 | 47.2 | 22.2 | 58.2 | 6.7 | - | - | - |
| MengziBERT (base) | 86.9 | 31.5 | 47.3 | 21.8 | 57.2 | 6.1 | - | - | - |
| RoBERTa (base) | 86.3 | 28.5 | 48.6 | 22.2 | 57.7 | 6.4 | - | - | - |
| RoBERTa (large) | 88.7 | 33.0 | 50.8 | 22.8 | 59.5 | 6.0 | - | - | - |
| BART (large) | 86.5 | 32.8 | 50.2 | 20.6 | 56.7 | 7.0 | 41.6 | 10.2 | 12.2 |
| mBART (CC25) | - | - | - | 23.5 | 23.1 | 15.0 | - | - | - |
| GPT2 | - | - | - | - | - | - | 37.3 | 13.5 | 13.5 |
| T5 | - | - | - | - | - | - | 35.0 | 12.2 | 16.1 |
| *GuoFeng Pretrained Models* | | | | | | | | | |
| RoBERTa (base) | 87.7 | 31.2 | 46.6 | 22.5 | 58.7 | 6.6 | - | - | - |
| RoBERTa (large) | **89.6** | **34.3** | **52.2** | 21.0 | 60.1 | 7.2 | - | - | - |
| BART (large) | 87.6 | 33.5 | 50.3 | 21.0 | **62.7** | 7.1 | **45.2** | **8.5** | **8.0** |
| mBART (CC25) | - | - | - | **24.0** | 25.1 | **15.7** | - | - | - |
| GPT2 | - | - | - | - | - | - | 40.4 | 9.6 | 9.1 |

Table 4: Performance of baseline models on GuoFeng benchmark. A similar table is presented on the online platform. **Bold** denotes the best result in each column. SI and ZPR are measured by F1 while MRC by accuracy. We report BLEU for NT, CCT, PT and TE, and PPL for others. Results using additional evaluation metrics are reported in Appendix §B.

across domains. However, most existing pretrained models are trained on datasets without discourse information (e.g. sentence level) or in mixed domains (e.g. Wikipedia and news). Considering that texts in literature domain contains rich discourse phenomena (De Beaugrande & Dressler, 1981), we construct a large-scale, in-domain and document-level datasets in Chinese and English. The data statistics are detailed in Table 14. To fill the gap, we follow Wang et al. (2022) to train the existing pretraining models (*coarse-grained pretraining*) on the document-level GuoFeng data in the literal domain (*fine-grained pretraining*) to model discourse phenomena. For each GuoFeng model, we use the existing pretrained models for weight initialization, and we further train the models on the GuoFeng data with the same loss. We limit the input length to 512 tokens for RoBERTa models and 1024 tokens for BART, mBART, and GPT models. The pretraining hyper-parameters details of the release Guofeng models can be found in Table 15.

## 4.2 MAIN RESULTS

Table 4 lists the results on the proposed benchmarks, where several observations can be made. Concerning the existing pretrained models, pretraining improves performance over plain models in all tasks, which is consistent with previous studies. These results validate that the proposed benchmarks are reasonable. Among the BERT variants with the base setting, AncientBERT trained on small-scale classical Chinese data outperforms other models on CCT and PT, demonstrating the necessity of bridging the domain gap. Enlarging the model capacity (e.g. from base to large setting) consistently improves performance.

Clearly, fine-grained pretraining on GuoFeng data outperforms their coarse-grained counterparts, demonstrating the effectiveness and necessity of modeling discourse information. The RoBERTa models work better on language understanding tasks, and the BART variants produce superior performances on the language translation and generation tasks.

| Model | Understanding | | | Translation | | | Generation | | |
|---|---|---|---|---|---|---|---|---|---|
| | SI | ZPR | MRC | NT | ACT | PT | TE | TI | TC |
| *Existing Pretrained Models* | | | | | | | | | |
| RoBERTa (large) | 0.1/0.5 | 0.4/0.4 | 0.3/0.2 | 0.7/0.7 | 0.8/0.8 | 0.5/0.6 | - | - | - |
| BART (large) | 0.2/0.4 | 0.3/0.4 | 0.3/0.2 | 0.7/0.7 | 0.7/0.8 | 0.5/0.6 | 0.8/0.7 | 0.6/0.2 | 0.5/0.6 |
| mBART (CC25) | - | - | - | 0.8/0.7 | 0.2/0.4 | 0.6/0.6 | - | - | - |
| *GuoFeng Pretrained Models* | | | | | | | | | |
| RoBERTa (large) | 0.1/0.5 | 0.4/0.4 | 0.4/0.2 | 0.7/0.7 | 0.8/0.8 | 0.6/0.6 | - | - | - |
| BART (large) | 0.2/0.5 | 0.4/0.4 | 0.4/0.2 | 0.8/0.7 | 0.8/0.8 | 0.6/0.6 | 0.8/0.7 | 0.6/0.4 | 0.5/0.7 |
| mBART (CC25) | - | - | - | 0.8/0.8 | 0.4/0.4 | 0.8/0.7 | - | - | - |

Table 5: Results of selected baseline models on GuoFeng diagnostic dataset. We assess models on their ability to rank the correct candidate higher than the incorrect one according to model score. We report accuracy (%) in terms of cohesion/coherence.

### 4.3 ANALYSIS

We evaluate three representative models on the diagnostic dataset: RoBERTa (large), BART (large), and mBART (CC25). Each model is fine-tuned on the training data of the corresponding downstream task (e.g. SI) and then tested on our diagnostic dataset. As illustrated in Table 5, GuoFeng pretrained models generally improve the cohesion and coherence accuracies over their coarse-grained counterparts, which reconfirms our claim that fine-grained pretraining on GuoFeng data helps model discourse information. Although the numbers are not comparable across tasks, we find that pretraining models on the understanding tasks generally perform worse on discourse modeling. One possible reason is that the understanding tasks are mostly classification tasks, whose signals may not be sufficient to guide models to learn discourse information.

## 5 RELATED WORK

Evaluation benchmarks are important for developing deep learning models, which enable comparison between different models and probe models for understanding of specific linguistic phenomena. Conneau & Kiela (2018) collected SentEval containing several sentence-level classification tasks to test the representational power of models. Closely related to this work, DiscoEval (Chen et al., 2019) extended these tasks to evaluate discourse-related knowledge in pretrained models. DiscoEval only evaluates sentence encoder with language understanding tasks in English. In contrast, we extend the tasks to a boarder range of NLP tasks, which can evaluate different types of models (e.g. encoder-based BERT, decoder-based GPT, and encoder-decoder based mBART). In addition, our benchmarks cover both Chinese and English.

GLUE (Wang et al., 2018) and SuperGLUE (Wang et al., 2019) included a wider variety of natural language understanding tasks, further examining the capabilities of the models and making the results comparable for multi-task learning. Followed researchers extend the benchmarks to other languages, such as CLUE (Xu et al., 2020) and LOT (Guan et al., 2022) in Chinese, and XGLUE (Liang et al., 2020) in multiple languages. While these works focus on evaluating inter-sentence information,[7] our benchmark evaluates intra-sentence discourse phenomena that cross sentences.

## 6 CONCLUSION

This paper introduces a GuoFeng benchmark for Chinese and/or English that can evaluate intra-sentence discourse properties across a diverse set of NLP tasks, covering understanding, translation, and generation. We also propose a diagnostic test suite that can examine whether the target models learn discourse knowledge for in-depth linguistic analysis. Extensive experiments demonstrate that fine-grained pretraining based on document-level training data consistently improves the modeling of discourse information. We offer the datasets, pretrained models, and leaderboards to facilitate research in this field.

---

[7]LOT (Guan et al., 2022) evaluates models' abilities to model long text but ignores discourse information.

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

## A    CONCEPT OF DISCOURSE

A discourse is an instance of language use whose type can be classified on the basis of such factors as grammatical and lexical choices and their distribution in main versus supportive materials, theme, style, and the framework of knowledge and expectations within which the addressee interprets the discourse (Elson & Pickett, 1983; Crystal, 1985; Hanks, 1987; Longacre, 1990). A discourse contains seven fundamental properties including cohesion, coherence, intentionality, acceptability, informatively, situationality and intertextuality (De Beaugrande & Dressler, 1981). Among them, *cohesion* and *coherence* have often been studied in discourse analysis (Sanders & Maat, 2006; Xiong et al., 2013).

### A.1    COHESION

Cohesion occurs whenever "the interpretation of some element in the discourse is dependent on that of another" (Halliday & Hasan, 1976). The referential cohesion (i.e. *anaphora* and *coreference*) and lexical cohesion (i.e. *repetition* and *collocation*) are commonly-used cohesive devices. The examples are shown in Figure 5.

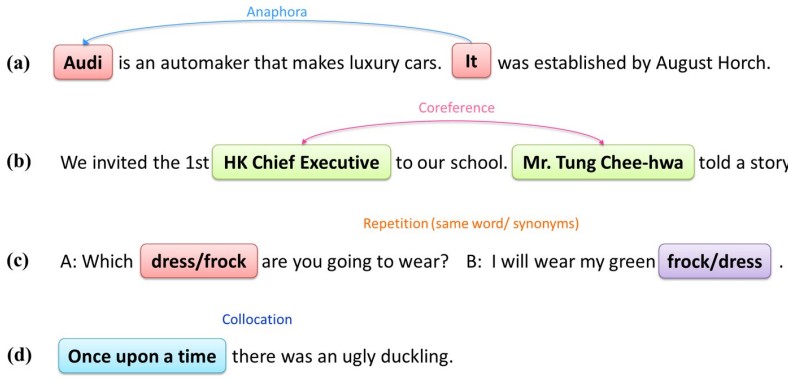

Figure 5: Examples of different cohesion devices.

**Anaphora**    It is the use of an expression whose interpretation depends specifically upon antecedent expression. The anaphoric (referring) term is called an anaphor. Sometimes anaphor may rely on the postcedent expression, and this phenomenon is called cataphora. As shown in Figure 5(a), the pronoun "It" is an anaphor, which points to the left toward its antecedent "Audi". Zero anaphora is a more complex case of anaphora. In pro-drop languages such as Chinese and Japanese, pronouns can be omitted to make the sentence compact yet comprehensible when the identity of the pronouns can be inferred from the context. These omissions may not be problems for our humans since we can easily recall the missing pronouns from the context.

**Coreference**    Two or more expressions (e.g. nouns) in a text refer to the same referent. As the referents point to persons or things in the real world, the coreference relation can exist independently of the context. As shown in Figure 5(b), the noun phrases "HK Chief Executive" and "Mr. Tung Chee-hwa" point to the same person, although their surfaces are totally different.

**Lexical Cohesion**    Lexical cohesion refers to the way related words are chosen to link elements of a text. The "repetition" indicates the linking between the same word, or synonyms, antonyms, etc. As shown in Figure 5(c), the synonyms "dress" and "frock" across two sentences are the repetition case. In the "collocation" form, related words are typically put together or tend to repeat the same meaning. For example, the phrase "once upon a time" in Figure 5(d) is a collocation case.

### A.2    COHERENCE

Coherence is created referentially, when different parts of a text refer to the same entities, and relationally, by means of coherence relations such as "Cause–Consequence" between different

discourse segments. The discourse structure such as RST (Rhetorical Structure Theory, (Mann & Thompson, 1988) is usually used to analyze the coherence of a text.

RST relations are applied recursively in a text until all units in that text are constituents in a predefined relation. As shown in Figure 6, the result of such analysis is that RST structure is typically represented as a tree, with one top-level relation that encompasses other relations at lower levels. There are a number of predefined relations such as "Attribution" (causality) and "Contrast" (adversative relation), and the leaves are presented as segments/parts of the text.[8]

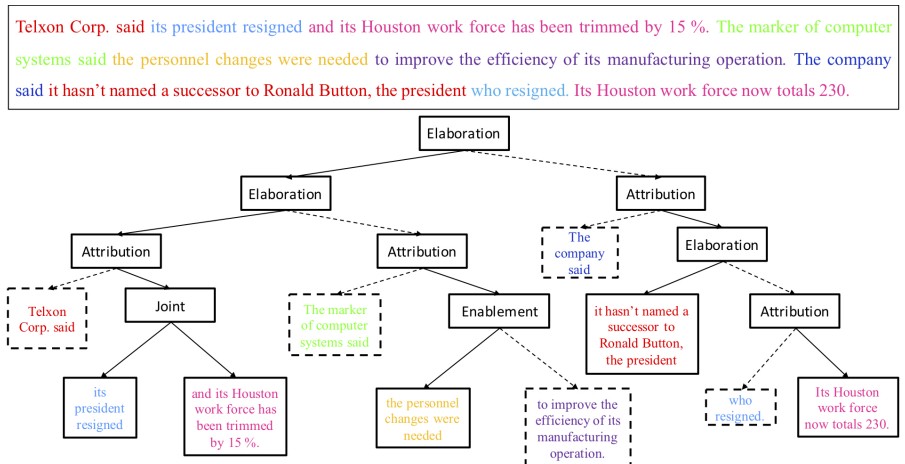

Figure 6: An example of coherence represented by RST tree.

# B  ADDITIONAL TASK AND RESULT DETAILS

The English translation of examples in Figure 2, 3 and 4 are listed in Table 6.

## B.1  LANGUAGE UNDERSTANDING TASKS

**SI**  The macro-averaged F1 and exact match (EM) can be used as the evaluation metrics following standard extractive machine reading comprehension tasks (e.g. (Rajpurkar et al., 2016)). The EM scores are reported in Table 8.

**ZPR**  Table 7 shows all Chinese pronouns with English translation. Chinese pronouns correspond to the personal pronouns in English, and the Chinese pronominal system is relatively simple as there is no inflection, conjugation, or case makers (Li & Thompson, 1989). Thus, there is no difference between subjective and objective pronouns (we call them "basic pronouns"). Besides, possessive and reflexive pronouns can be generated by adding some particle or modifier based on the basic pronouns.

The (zero) anaphora resolution is an alternative task on discourse understanding, which aims to identify the antecedent of a referential (zero) pronoun (Kong & Zhou, 2010; Mitkov, 2014). However, we did not consider this task for two reasons: (1) more than 50% zero pronouns are non-anaphoric which can not be modelled in the resolution task (Rao et al., 2015); (2) different from previous benchmarks such as OntoNotes and CLUEWSC2020 which mainly focus on explicit pronouns, while ZPR considers implicit pronouns which are complementary to each other. We follow common practice to use micro F1 score as the standard evaluation metric. And we also report precision and recall in Table 8.

**MRC**  Considering the average length of texts, the Haihua2021 corpus is also more challenging than $C^3$ (i.e. the length ratio is 753:117).

---

[8]http://www.sfu.ca/rst/index.html.

| Task | Discourse Context | Task Description |
|------|-------------------|-----------------|
| | *Figure 2* | |
| SI | Xing Jiu'an followed Mu Qing into the car and sat in the co-pilot position.
"Are you in a bad mood?" Mu Qing asked.
"Um, yes." | Inp: "Um, yes."
Out: Speaker=Xing Jiu'an |
| ZPR | A: Phoebe would love to buy a TV.
B: Joey won't let ∅ buy ∅?
A: Yes. | Inp: B: Joey won't let ∅ buy ∅?
Out: B: Joey won't let her buy it? |
| MRC | The little princess climbed out of the castle window while her mother was sleeping.
She climbed down the south wall and slipped out.
Finally ∅ walked into the forest without telegraph poles. | Inp: Where did the little princess go after she escaped?
(A) South Wall; (B) Forest; (C) Castle; (D) Mountain.
Out: Answer=(B) Forest |
| | *Figure 3* | |
| NT | King Ding sat on the side,
smiling as he looked at Qing Shuang's astounded thoughts.
∅ mind had already flown to a faraway place. | Inp: ∅ mind had already flown to a faraway place.
Out: – |
| CCT | ©, when she is playing Xiao, not only can her beautiful face remain as usual, but also her charm increases.
Why?
© ∅ is playing, ∅ fingers press the holes on the flute, and in this way, ∅ tender and slim fingers will seem to be slimmer and fairer.
©, when shrinking ∅ month to blow, ∅ mouth appears to be smaller. | Inp: ©, when shrinking ∅ month to blow, ∅ mouth appears to be smaller.
Out: Besides, when shrinking her month to blow, her mouth appears to be smaller. |
| PT | I ask your lad beneath a tree.
"My master's gone for herbs, " says he,
"Amid the hills I know not where,
For clouds have veiled them here and there. " | Inp: I ask your lad beneath a tree.
Out: – |
| | *Figure 4* | |
| TE | – | – |
| TI | Mu Xiaoxiao looked at his back aggrieved, why did it suddenly change like this?
She was inexplicably trained for a while, which made her feel bad.
When she got to class S, she was lying on the table and was sullen. | Inp: Mu Xiaoxiao looked at his back aggrieved, why did it suddenly change like this? [x] [x] [x] ... When she got to class S, she was lying on the table and was sullen.
Out: She was inexplicably trained for a while, which made her feel bad. |
| TC | Chen Xu was hungry and cold. He used a small gas stove to cook a pot of noodles.
The two gathered around the pot and devoured everything.
After they ate the noodles, they felt alive. | Inp: Chen Xu was hungry and cold. [x] [x] [x] ...
Out: The two gathered around the pot and devoured everything. After they ate the noodles, they felt alive. |

Table 6: English translations of examples in Figure 2, 3 and 4. Some are literal translations in order to map discourse phenomena into English language.

## B.2 LANGUAGE TRANSLATION TASKS

There are a number of evaluation metrics for measuring general performance of MT systems. BLEU is the most widely-used one, which measures the precision of n-grams of the MT output compared to the reference, weighted by a brevity penalty to punish overly short translations (Papineni et al., 2002). TER is an error metric for machine translation that messures the number of edits required to change a system output into one of the references (Snover et al., 2006). METEOR incorporates semantic information by calculating either exact match, stem match, or synonymy match (Banerjee & Lavie, 2005). Furthermore, COMET is a neural framework for training multilingual MT evaluation models which obtains new SOTA levels of correlation with human judgements (Rei et al., 2020). We report TER, METEOR and COMET in Table 9.

**NT (Novel Translation)**    We crawl 45,134 chapters in 152 books from web fiction websites, covering 14 genres such as fantasy science and romance. We manually align them into parallel corpus with document boundary. The copyright owner has agreed to open the data to research community

| Form | Subject | Object | Possessive adjective | Possessive | Reflexive |
|---|---|---|---|---|---|
| 1st SG | 我 (I) | 我 (me) | 我的 (my) | 我的 (mine) | 我自己的 (myself) |
| 2nd SG | 你 (you) | 你 (you) | 你的 (your) | 你的 (yours) | 你自己的 (yourself) |
| 3rd SGM | 他 (he) | 他 (him) | 他的 (his) | 他的 (his) | 他自己的 (himself) |
| 3rd SGF | 她 (she) | 她 (her) | 她的 (her) | 她的 (hers) | 她自己的 (herself) |
| 3rd SGN | 它 (it) | 它 (me) | 它的 (its) | 它的 (its) | 它自己的 (itself) |
| 1st PL | 我们 (we) | 我们 (us) | 你们的 (your) | 你们的 (yours) | 你们自己的 (yourselves) |
| 2nd PL | 你们 (you) | 你们 (you) | 我们的 (our) | 我们的 (ours) | 我们自己的 (ourselves) |
| 3rd PLM | 他们 (they) | 他们 (them) | 他们的 (their) | 他们的 (theirs) | 他们自己的 (themselves) |
| 3rd PLF | 她们 (they) | 她们 (them) | 她们的 (their) | 她们的 (theirs) | 她们自己的 (themselves) |
| 3rd PLN | 它们 (they) | 它们 (them) | 它们的 (their) | 它们的 (theirs) | 它们自己的 (themselves) |

Table 7: Chinese-English pronouns with corresponding forms. The pronoun types are short for: person = 1st, 2nd, 3rd, singular = SG, plural = PL, male = M, female = F and neutral = N.

| Model | SI | ZPR | |
|---|---|---|---|
| | Exact Match$^\uparrow$ | Precision$^\uparrow$ | Recall$^\uparrow$ |
| *Plain Models* | | | |
| Transformer (base) | 0.3 | 10.2 | 11.5 |
| Transformer (big) | 0.1 | 10.5 | 11.9 |
| *Existing Pretrained Models* | | | |
| BERT (base) | 81.9 | 26.1 | 31.0 |
| AnchiBERT | 76.9 | 22.1 | 24.6 |
| MengziBERT | 84.0 | 36.6 | 29.6 |
| RoBERTa (base) | 83.4 | 29.0 | 29.9 |
| RoBERTa (large) | 85.9 | **39.3** | 28.7 |
| BART (large) | 83.7 | 38.3 | 30.2 |
| *GuoFeng Pretrained Models* | | | |
| RoBERTa (base) | 85.2 | 32.0 | 30.6 |
| RoBERTa (large) | **87.2** | 38.7 | **30.8** |
| BART (large) | 84.6 | 39.0 | 30.5 |

Table 8: More results on **understanding tasks** using additional evaluation metrics, including Exact Match, Precision, and Recall. This is complementary to Table 4.

under the Open Data Commons Open Database License (ODbL). Thus, researchers can use the dataset for non-commercial research purposes and follow the principle of fair use (e.g. CC BY).

### B.3 LANGUAGE GENERATION TASKS

There are a number of automatic evaluation metrics for measuring the quality of generated texts. We use two groups of metrics: (1) Reference-based scores BLEU (Papineni et al., 2002) and BERTScore (Zhang et al., 2019a), which measure the lexical and semantic similarities between the generated texts and the ground-truth references respectively. Note that, for open-ended text generation tasks such as TI and TC, reference-based metrics are less reliable because the generated text could be of high quality but different from the ground-truth reference. How to accurately measure the performance of open-ended text generation is still an open question and is beyond the scope of this paper. (2) Dist-$n$ scores (Li et al., 2016) calculate the ratio of distinct n-grams in generated text to evaluate lexical diversity. We report Dist-2 and Dist-4 as well. Additional results are shown in Table 10.

**TE** To better understand the task of TE, we present some examples in Table 11. In order to retain the coherence and meaning of the source document, the expanded parts in the target text tends to be modifier phrases or clauses. In general, the expanded contents in our task are mainly summarized into 5 categories, shown in Table 12. To evaluate the TE models, we use two metrics: BLEU and PPL.

| Model | NT | | | CCT | | | PT | | |
|---|---|---|---|---|---|---|---|---|---|
| | TER$\downarrow$ | MET.$\uparrow$ | COM.$\uparrow$ | TER$\downarrow$ | MET.$\uparrow$ | COM.$\uparrow$ | TER$\downarrow$ | MET.$\uparrow$ | COM.$\uparrow$ |
| *Plain Models* | | | | | | | | | |
| Transformer (base) | 70.0 | 24.0 | 0.10 | 70.4 | - | 0.65 | 104.5 | 10.3 | -0.82 |
| Transformer (big) | 69.2 | 24.5 | 0.14 | 77.5 | - | 0.65 | 103.6 | 11.1 | -0.78 |
| *Existing Pretrained Models* | | | | | | | | | |
| BERT (base) | 69.6 | 24.0 | 0.10 | 27.6 | - | 0.65 | 97.0 | 13.3 | -0.68 |
| AnchiBERT | 69.5 | 24.3 | 0.11 | 27.1 | - | 0.67 | 91.4 | 13.4 | -0.66 |
| MengziBERT | 69.8 | 23.9 | 0.10 | 27.7 | - | 0.67 | 96.4 | 11.8 | -0.71 |
| RoBERTa (base) | 69.7 | 24.0 | 0.13 | 27.2 | - | 0.65 | 96.0 | 12.2 | -0.71 |
| RoBERTa (large) | 68.9 | 25.0 | 0.15 | 27.0 | - | 0.68 | 97.5 | 11.9 | -0.75 |
| BART (large) | 70.9 | 24.6 | 0.12 | 28.2 | - | 0.65 | 90.5 | 13.5 | -0.74 |
| mBART(CC25) | **64.7** | 26.9 | 0.28 | - | - | 0.23 | **72.4** | 21.7 | -0.14 |
| *GuoFeng Pretrained Models* | | | | | | | | | |
| RoBERTa (base) | 69.7 | 24.2 | 0.14 | 27.3 | - | 0.67 | 96.0 | 12.3 | -0.70 |
| RoBERTa (large) | 70.0 | 24.9 | 0.15 | 28.3 | - | 0.67 | 95.9 | 13.0 | -0.74 |
| BART (large) | 70.0 | 24.8 | 0.12 | **25.6** | - | **0.70** | 90.4 | 13.3 | -0.74 |
| mBART (large) | 64.8 | **27.0** | **0.30** | - | - | 0.23 | 73.3 | **22.0** | **-0.11** |

Table 9: More results on **translation tasks** using additional evaluation metrics, including TER, METEOR and COMET. This is complementary to Table 4.

| Model | TE | TI$\uparrow$ | | | | TC$\uparrow$ | | | |
|---|---|---|---|---|---|---|---|---|---|
| | PPL$\downarrow$ | BLEU | BERTscore | Dist-2 | Dist-4 | BLEU | BERTscore | Dist-2 | Dist-4 |
| *Existing Pretrained Models* | | | | | | | | | |
| BART (large) | 48.0 | 3.7 | 62.2 | **0.2** | 0.6 | 2.7 | 60.3 | **0.1** | 0.4 |
| GPT2 | 50.7 | 1.6 | 59.4 | **0.2** | 0.5 | 2.1 | 57.6 | 0.0 | 0.2 |
| T5 | 54.8 | 3.3 | 61.1 | 0.1 | 0.5 | 2.2 | 57.3 | 0.0 | 0.1 |
| *GuoFeng Pretrained Models* | | | | | | | | | |
| BART (large) | **34.0** | **3.9** | **62.5** | **0.2** | **0.7** | 4.2 | **61.0** | 0.0 | **0.6** |
| GPT2 | 45.0 | 2.2 | 60.2 | **0.2** | **0.7** | **4.7** | 60.2 | **0.1** | 0.5 |

Table 10: More results on **generation tasks** using additional evaluation metrics, including BLEU, BERTscore, Dist-2 and Dist-4. This is complementary to Table 4.

**TI** To evaluate different models, we take the following automatic metrics: Perplexity (PPL), BLEU (Papineni et al., 2002), BertScore (Zhang et al., 2019a) and diversity scores (Dist-2/4) (Li et al., 2016). We report degree of diversity by calculating the ratio of distinct 2-grams/4-grams in generated text.

**TC** To evaluate different models, we take the following automatic metrics: Perplexity (PPL), BLEU, BertScore and diversity scores.

## C  ADDITIONAL EXPERIMENT DETAILS

We evaluate the following public pretrained models on GuoFeng Benchmark and Test Suite:

- **BERT (base)**: we use the base model (12 layer encoder, hidden size 768, vocabulary size 21128) published by Devlin et al. (2019), which was pretrained on Chinese Wikipedia dump of about 0.4 billion tokens using the losses of mask language model (MLM) and next sentence prediction. [9]

- **RoBERTa (base)**: Cui et al. (2020) a model with the same architecture of BERT (base) except it uses whole word masking and is trained on additional 5 billion tokens with only MLM pretrained task. This model uses BERT (base) as the initial weight. [10]

---

[9] https://huggingface.co/bert-base-chinese.

[10] https://huggingface.co/hfl/chinese-roberta-wwm-ext/tree/main.

| Source 1 | In 1823 James Monroe proclaimed the doctrine. The United States was an infant, threatened by European actions. |
|---|---|
| Target 1 | To this effect, in 1823 President James Monroe proclaimed the doctrine that still bears his name. The United States at this time was an infant, weak country, threatened by European actions. |
| Source 2 | First was the rule. Political democracies have not been institutionalized in parts of Latin America. No democratic regime had lasted half a century. |
| Target 2 | First was the weakness of democratic rule. Political democracies even now still have not been firmly institutionalized in parts of Latin America. In the past no democratic regime had lasted half a century. |
| Source 3 | The peasant was sentenced to death, and was to be rolled into the water. He was led forth, and a priest was brought. |
| Target 3 | The innocent little peasant was unanimously sentenced to death, and was to be rolled into the water, in a barrel pierced full of holes. He was led forth, and a priest was brought who was to say a mass for his soul. |

Table 11: Three examples to illustrate the task of TE, where the blue span in Target are expanded content generated based on the source input as context.

| Expanded Content Type | Exemplar Spans |
|---|---|
| (1) adjective (phrase) | innocent little |
| (2) adverb (phrase) | firmly, even now still, unanimously |
| (3) noun (phrase) | President, weak country, weakness of democratic |
| (4) prepositional phrase | To this effect, In the past, at this time, in a barrel pierced full of holes |
| (5) attributive clause | that still bears his name, who was to say a mass for his sou |

Table 12: The expansion types in TE task are summarized. All the exemplar spans are highlighted in texts in Table 11.

- **RoBERTa (large)**: Cui et al. (2020) the large model size of RoBERTa model (24 layer encoder, hidden size 1024, vocabulary size 21128) This model has the same training procedure of RoBERTa-wwm-ext (base). This model is trained from scratch. [11]

- **AnchiBERT**: Tian et al. (2021) a model continues pretraining based on the BERT (base) model with the 39.5M ancient Chinese tokens. It uses the same tokenizer and other techniques as BERT-base. [12]

- **MengziBERT**: Zhang et al. (2021) a model initial on the RoBERTa (base) (Liu et al., 2019b) with special-designed objectives. [13]

- **BART (large)**: Shao et al. (2021) train a large model (12 layer encoder and 12 layer decoder, hidden size 1024, vocabulary size 21128) with denoising auto-encoding (DAE) objective. This model is trained on the open source large-scale raw text, Chinese Wikipedia, and a part of WuDaoCorpus. The training data contains 200GB cleaned text ranging from different domains. [14]

- **mBART (CC25)**: Pires et al. (2019) use a large model (12 layer encoder and 12 layer decoder, hidden size 1024, vocabulary size 250,000), trained with 25 language web corpus. This model is trained from scratch. [15].

- **GPT2**: Zhao et al. (2019) train a 12-layer decoder-only Transformers and its vocabulary is size 21,128. This model is trained with the CLUECorpusSmall corpus. [16].

---

[11] https://huggingface.co/hfl/chinese-roberta-wwm-ext.

[12] https://github.com/ttzHome/AnchiBERT.

[13] https://huggingface.co/Langboat/mengzi-bert-base.

[14] https://huggingface.co/fnlp/bart-base-chinese.

[15] https://dl.fbaipublicfiles.com/fairseq/models/mbart/mbart.cc25.v2.tar.gz

[16] https://github.com/CLUEbenchmark/CLUECorpus2020.

- **T5**: For Chinese langauge, Zhao et al. (2019) train a 12-layer encoder-decoder Transformers and its vocabulary is size 21,128. This model is trained with the CLUECorpusSmall corpus. [17]. For English langauge, Raffel et al. (2020) provides a transformer model (24 layer encoder and 24 layer decoder, hidden size 1024, vocabulary size 32,000), trained with unified generation task on C4 dataset. [18]

| Task | Batch Size | Max Length | Epoch | Learning Rate |
|------|-----------|-----------|-------|---------------|
| SI | 64 | 512 | 5 | 3e-5 |
| ZPR | 5 | 512 | 40 | 5e-6 |
| MRC | 6 | 512 | 10 | 2e-5 |
| NT | 3K token | 1024 | 30K step | 1e-4 |
| ACT | 3K token | 1024 | 30K step | 1e-4 |
| PT | 3K token | 1024 | 30K step | 1e-5 |
| TE | 32 | 512 | 3 | 2e-4 |
| TI | 24 | 64 | 3 | 2e-5 |
| TC | 24 | 512 | 8 | 2e-5 |

Table 13: A summary of hyper-parameter for fine-tuning downstream tasks.

| Category | Genre | Size | | | Description |
|----------|-------|------------|------------|-------------------|-------------|
| | | # Document | # Sentence | # Character/Word | |
| | | *Chinese Language* | | | |
| Electronic | Novel | 91,620,211 | 1,169,127,191 | 58,639,454,317 | Web Fiction |
| Modernist | Classical | 38,495,887 | 490,733,235 | 24,613,514,541 | Masterpiece |
| | Book | 324,912 | 4,141,874 | 155,189,807 | Publication |
| Ancient | Poetry | 378,323 | 1,495,466 | 31,746,541 | Shi, Ci, Qu, Fu |
| | Couplet | 8,979,186 | 8,979,186 | 192,214,600 | Antithetical Couplet |
| | Classical | 1,011 | 1,947,136 | 53,721,504 | Ancient Text |
| Others | Lyrics | 452,715 | 4,952,039 | 165,338,679 | World's Songs |
| | Screenplay | 5,213 | 10,426,213 | 156,390,000 | Movie Script |
| | Movie | 66,050 | 24,108,241 | 642,392,397 | Movie Subtitle |
| | Dialogue | 3,642 | 1,653,469 | 49,406,618 | Talk, Message |
| Total | | 140,327,150 | 1,717,564,050 | 84,699,369,004 | |
| | | *English Language* | | | |
| Electronic | Novel | 33,156,134 | 422,757,234 | 26,777,401,794 | Web Fiction |
| Modernist | Classical | 3,104,507 | 39,593,119 | 2,507,247,359 | Masterpiece |
| | Book | 324,912 | 4,162,821 | 78,695,499 | Publication |
| Ancient | Poetry | 2,269 | 21,456 | 148,222 | World's Poetry |
| Others | Lyrics | 3,088,688 | 110,268,328 | 632,820,393 | World's Songs |
| | Movie Script | 2,826 | 12,534,815 | 67,433,609 | Movie Script |
| | Movie | 155,670 | 56,819,567 | 315,189,001 | Movie Subtitle |
| | Dialogue | 9,191 | 4,172,736 | 27,208,957 | Talk, Message |
| Total | | 39,844,197 | 650,330,076 | 30,406,144,834 | |

Table 14: Statistics of data for GuoFeng pretraining. All data are extracted from literature texts with discourse context. We count number of characters in Chinese and number of words in English.

---

[17] https://github.com/CLUEbenchmark/CLUECorpus2020.
[18] https://github.com/google-research/text-to-text-transfer-transformer.

| Model | RoBERTa | GPT2 | BART | mBART |
|---|---|---|---|---|
| Tokenization | BERTtok. | BERTtok. | BERTtok. | SentPiece |
| Optimizer | Adam | Adam | Adam | Adam |
| Masking | word | - | word | word |
| Vocabulary Size | 21128 | 21131 | 21128 | 250000 |
| Learning Rate | 3e-4 | 3e-4 | 3e-4 | 3e-4 |
| Batch Size | 4K | 4K | 4K | 4K |
| Training Step | 1M | 1M | 1M | 1M |
| Max Length | 512 | 512 | 512 | 1024 |
| Layer | 12/24 | 36 | 24 | 12/24 |
| Head | 12/16 | 36 | 16 | 12/16 |
| Total Param. | 110m/340m | 737M | 477M | 669M |

Table 15: The summary of hyper-parameters used for Guofeng pretrained models.

