# OpenReview forum: "GuoFeng: A Discourse-aware Evaluation Benchmark for Language Understanding, Translation and Generation"
_ICLR.cc/2023/Conference — Submitted to ICLR 2023_

### Official Review · Reviewer_zDLD · 2022-10-27

**Confidence:** 3
**Correctness:** 3
**Technical Novelty And Significance:** 4
**Empirical Novelty And Significance:** 3
**Recommendation:** 8

**Clarity, Quality, Novelty And Reproducibility:**

The paper is exceptionally clear, of high quality, and while it is a resource paper so it does not contribute novel modelling, it does offer a very thorough account of discourse in natural language understanding, translation, and generation.

**Details Of Ethics Concerns:**

None.

**Strength And Weaknesses:**

Strengths:
- Potentially a very useful resource for discourse-aware evaluation across tasks.
- Very thorough description of the resources, including in-depth background in the appendix.
- Considerable experiment breadth.
- High-quality writeup.

Weaknesses:
- The paper does not reflect thoroughly on the linguistic bias introduced by choosing Chinese and English as the focus languages. There should be a Limitations section that addresses how the proposed approach would scale in low-resource scenarios or if extending the breadth of linguistic coverage in the datasets. What would it take to introduce another language? What if that language is low-resource? How would that support the findings? Can we simulate that with the existing data to an extent?
- Experiment breadth is not backed with experiment depth, at least in the main part of the paper, as the Main Results and Analysis are rather superficial. This is a minor concern since the main focus of the paper is to provide the resource.

**Summary Of The Paper:**

The paper proposes a novel set of benchmark resources for discourse-aware evaluation of language undersatnding, translation, and generation, focusing on Chinese and English languages. The release includes benchmarking data, diagnostic test suite, pretraining corpora and pretrained models. The benchmark is applied on relevant models across a wide variety of tasks, showing benefits of document-level pretraining in Transformer architectures when it comes to discourse.

**Summary Of The Review:**

Contained above.

---

> ### Author Response · Authors · 2022-11-18
> **Response to Reviewer zDLD**
>
> **Q1. Add the limitation section to discuss the potential bias and solutions.**
>
> We will add this section to clarify the linguistic bias and possible solutions.
>
> The discourse phenomena may vary across languages in terms of element form, occurrence frequency and type distribution etc. [1]. Our work mainly focuses on Chinese and English due to the lack of linguistic experts in other languages. Another reason is the high cost, where we totally spent \$1,000,000 US dollars for data collection, manual annotation, and model training.
>
> However, this may result in linguistic bias because English and Chinese can not cover all discourse properties across different languages. For example, grammatical gender significantly affects our discourse-aware tasks (e.g. pronoun recovery and translation). This is not a problem for English and Chinese languages but for French and Spanish. In the text: *Audi is an automaker that makes luxury cars*. *It was established by August Horch.* The pronoun “it” could be translated into “il” (masculine singular subject pronoun), “elle” (feminine singular subject pronoun) or “cela” (demonstrative pronoun) according to its antecedent gender “Audi”. Therefore, more discourse-aware datasets in different languages can provide insights into language-specific discourse modeling. One aim of our work is attracting more attention to this research topic, which will stimulate further contributions to building multilingual resources. Here we provide alternative methods to extend our datasets to other languages:
>
> -   For high-resourced languages, we can directly use document-level MT systems to construct these discourse-aware datasets in corresponding languages. To improve the quality of resulting data, native speakers can further post-edit translation outputs.
> -   For low-resourced languages, the performance of MT systems is far from practical use. First, researchers can simulate our methods to collect literature-domain data. The linguistic experts in the native language can adapt our annotation guidelines to label discourse elements task datasets or test suites.
>
> **Q2. Experiment breadth is not backed with experiment depth, however, this is a minor concern since the main focus of the paper is to provide the resource.**
>
> Thanks for the positive comments. GuoFeng is a comprehensive benchmark covering various NLP tasks, which require different discourse information to accomplish the task goal. Experimental results on the proposed benchmark show that existing pre-trained models lack the ability of modeling discourse: (1) no single model performs best in all tasks that require different types of discourse information; (2) GuoFeng models with fine-grained pretraining on document-level data consistently outperform their vanilla counterparts.
>
> Further analyses on the GuoFeng diagnostic test suit confirm our claim: existing pre-trained model can be easily attacked by simply modifying a single token that plays a discourse role. GuoFeng models can alleviate this problem, while there is still a large improvement space, indicating the necessity of more advanced techniques of modeling discourse information for pre-trained models.
>
> [1] Exploration of Inter- and Intralingual Variation of Discourse Phenomena. Ekaterina Lapshinova-Koltunski. Proceedings of the Second Workshop on Discourse in Machine Translation (DiscoMT). 2015.

---

### Official Review · Reviewer_CWMG · 2022-10-28

**Confidence:** 4
**Correctness:** 3
**Technical Novelty And Significance:** 2
**Empirical Novelty And Significance:** 2
**Recommendation:** 3

**Clarity, Quality, Novelty And Reproducibility:**

Reproducibility is problematic given that no quality control/measurement is described.

Clarity is an issue in several places.

**Strength And Weaknesses:**

Strengths: data set construction is valuable, and the goals of the work seem worthwhile.

Weaknesses: first, there is no attempt to evaluate the quality of the resulting data, either through expert evaluations or inter-annotator consistency (I think the former is the only way to really check for this). Second, many of the details of the paper are unclear. Third, many claims are made without sufficent support.

**Summary Of The Paper:**

The paper describes a new collection of datasets, the goal being to analyze discourse phenomena.

**Summary Of The Review:**

Overall, this is worthwhile work, but there are a few issues:

* the lack of any measures of quality of these datasets is problematic. I would be hesitant to use this data without such measures

* Clarity is an issue in several places. what does "aims to recover omitted pronouns in terms of position and form, according to its anaphora information in the given sentence" mean?, for example

* I think the argument that these are "discourse level" tasks is a stretch in several cases. For example, several of the datasets are conventional translation datasets.

Overall I think the goal to release data is valuable, but the paper needs to be improved.

---

> ### Author Response · Authors · 2022-11-18
> **Response to Reviewer CWMG (PART II)**
>
> **Q4. Please specify some weaknesses or questions.**
>
> Some comments are not specific enough. For example:
>
> -   "many of the details of the paper are unclear": which part? If it points to our settings, please check the answer of Response Q1\~Q2 in Reviewer eMwY.
> -   "many claims are made without sufficent support": which part?
> -   "clarity is an issue in several places": If it points to three questions in Summary, we have carefully answered them.
>
> Can you give more details about the comments? Thanks.
>
> **[Reference]**
>
> [1] Aya A Mitani, Phoebe E Freer, and Kerrie P Nelson. 2017. Summary measures of agreement and association between many raters' ordinal classifications. Annals of epidemiology.
>
> [2] Julia Kreutzer, Joshua Uyheng, Stefan Riezler. Reliability and Learnability of Human Bandit Feedback for Sequence-to-Sequence Reinforcement Learning. ACL 2018.
>
> [3] Rob Voigt and Dan Jurafsky. Towards a Literary Machine Translation: The Role of Referential Cohesion. ACL 2012.

---

> ### Author Response · Authors · 2022-11-18
> **Response to Reviewer CWMG (PART I)**
>
> **Q1. About evaluating quality of the resulting data through expert evaluations or inter-annotator consistency.**
>
> We ask professional linguistics and translators to evaluate 200 instances sampled from training sets and the whole testsets for each task. In general, all datasets are of high quality.
>
> 1.  **Datasets for understanding tasks**. As these data require manual annotation, the experts follow [1] to measure the inter-annotator agreement by calculating Cohen’s kappa score.
>
> | **Task/Datasets** | **Agreements** |
> |-------------------|----------------|
> | SI                | 0.76           |
> | ZPR               | 0.91           |
> | MRC               | 0.97           |
>
> 2.  **Datasets for translation tasks**. These data are collected from Internet and manually processed into parallel data. The experts measure fluency and adequacy, which are two typical issues of translation. We first compute average scores as data quality and then follow [2] to calculate Krippendorff's α as inter-annotator agreements on scoring.
>
> | **Task/Datasets** | **Fluency (1\~5)** | **Adequacy (1\~5)** | **Agreements**       |
> |-------------------|--------------------|---------------------|----------------------|
> | NT                | 4.9                | 4.7                 | flu. 0.60 adeq. 0.78 |
> | CCT               | 4.9                | 4.9                 | flu. 0.65 adeq. 0.55 |
> | PT                | 4.7                | 4.4                 | flu. 0.63 adeq. 0.69 |
>
> 3.  **Datasets for generation tasks**. These data are monolingual texts collected from books. The native speakers measure fluency and coherence. The rest of the processing is done in the same way as evaluation for translation datasets mentioned above.
>
> | **Task/Datasets** | **Fluency (1\~5)** | **Coherence (1\~5)** | **Agreements**        |
> |-------------------|--------------------|----------------------|-----------------------|
> | TE                | 4.0                | 4.1                  | flu. 0.51 coher. 0.51 |
> | TI                | 4.3                | 4.4                  | flu. 0.63 coher. 0.55 |
> | TC                | 4.3                | 4.4                  | flu. 0.63 coher. 0.55 |
>
> **Q2. What does "aims to recover omitted pronouns in terms of position and form, according to its anaphora information in the given sentence" mean?**
>
> Taking a Chinese text for example, words in brackets are omitted pronouns that are invisible in input. ZPR models need to recover them (e.g. 它) according to discourse information (e.g. its antecedent 小狗).
>
> Example:
>
> A: (你) 喜欢 这只 **小狗** 吗？ B: 是的，(我) 很喜欢 **(它)**，谢谢 (你)。
>
> A: Do you like this puppy? B: Yes, I like it. Thank you.
>
> **Q3. The "discourse level" tasks is a stretch in several cases. e.g. several of the datasets are conventional translation datasets.**
>
> Our translation datasets are from literuture domain, which are the first Chinese-English datasets in this domain. The literature datasets differ from existing discourse-level datasets (e.g. LDC in news domain, OpenSub in movie subtitle domain) at discourse richness and translation difficulty:
>
> 1.  **Discourse Richness**. Taking the zero anaphora for example, we calculate the entropy of zero pronoun types and frequency of the phenomenon across discourse-level datasets in five domains. Although texts in government news contain the most frequent phenomenon, the first person ZPs occupy a large proportion. Besides, the phenomenon in other domains is very sparse. These data may lead to severe bias in model behavior and performance evaluation. The literature-domain data (e.g. novel, classical Chinese and poem) with high frequency and diversity is an ideal discourse-level dataset.
>
> |     Domain            | Entropy of ZP Distribution  | Frequency of ZP Sentence |
> |-----------------|-----------------------------|--------------------------|
> | **Literature**      | **3.71**                    | 0.47                     |
> | Movie Subtitle  | 3.22                        | 0.36                     |
> | Q&A Forum       | 2.49                        | 0.44                     |
> | Government News | 1.37                        | **0.54**                 |
> | General News    | 0.81                        | 0.10                     |
>
> 2.  **Translation Difficulty**. Taking previous study [3] for example, it examines how referential cohesion is expressed in literary and non-literary texts and how this cohesion affects machine translation. They found that (1) literary texts use more dense reference chains to express greater referential cohesion than news. (2) advanced MT systems built with newswire texts in mind, will be less successful at conveying cohesion for literary texts than for news.

---

### Official Review · Reviewer_eMwY · 2022-10-29

**Confidence:** 4
**Correctness:** 3
**Technical Novelty And Significance:** 3
**Empirical Novelty And Significance:** 2
**Recommendation:** 5

**Clarity, Quality, Novelty And Reproducibility:**

* Clarity is certainly an issue, see the weaknesses.

* Reproducibility is also an issue due to the lack of human annotation details and the guidelines for human annotators.

**Strength And Weaknesses:**

Strengths:

* The datasets are valuable for the document-level NLP research, which need to address the coherence and cohesion across clauses and sentences.
* The experiments provide baseline results for future research.

Weaknesses:
* I have difficulties to fully understand the settings of the 9 tasks. How to evaluate each task?
     * SI: are the speakers always mentioned before the utterance of interest? What are the expected outputs if there are multiple utterances with quotation marks?
     * ZPR: What are Chinese-English movie subtitles? Are they Chinese movies with English subtitles, English movies with Chinese subtitles, or both?
     * NT: Is eclipsis not important for novel translation? How about concept consistency in scientice fictions?
     * CCT: do you mean traditional Chinese? Novels written in traditional Chinese?
     * PT: what's the target language? English?
     * How to evaluate the generation tasks if there are multiple plausible answers?
* What is the inter-annotator agreements for the tasks requiring human annotations?
* What are the inter-annotator agreements for the discourse-aware test suite? How are the annotators instructed to construct adversarial examples or the noises are added by algorithms? What kind of noises?
* Why coreference resolution models are not considered as part of or as baselines of applicable tasks?
* It is not clear to me what are the new challenges arising from the benchmark?

**Summary Of The Paper:**

This paper presents the GuoFeng benchmark consisting of i) 9 NLP tasks, ii) a contrastive testing set, iii) a large scale training data. The aim is to evaluate the abilities of models for cohesion and coherence across clauses and sentences. The authors also conduct experiments with transformer models using various settings, including randomly initialized ones, pre-trained models, and pre-trained models on the GuoFeng training data.

**Summary Of The Review:**

Overall, this paper can benefit from a major revision, although the resources are interesting for the community. More insights from the experiments are expected, not just model A is better than model B, expecially about what are the new challenges that are not covered by the prior works, evidenced by the experimental results.

---

> ### Author Response · Authors · 2022-11-18
> **Response to Reviewer eMwY (PART IV)**
>
> **Inter-annotator agreements of test suite**:
>
> We employed two linguistic experts to build the test suite. For tagging, we calculated the average pairwise Cohen’s kappa score [2]. For modification, we ask one annotator to vote accept/reject the modification from the other annotator.
>
> | **Task**          | **Cohesion Tagging** | **Cohesion Modification** | **Coherence Tagging** | **Coherence Modification** |
> |-------------------|----------------------|---------------------------|-----------------------|----------------------------|
> | Understanding | 90.6                 | 71.6                      | 82.3                  | 55.2                       |
> | Translation   | 89.1                 | 67.2                      | 82.2                  | 54.1                       |
> | Generation    | 92.6                 | 61.8                      | 79.9                  | 56.7                       |
>
> **Q5. Why coreference resolution models are not considered as part of or as baselines of applicable tasks?**
>
> In the preliminary experiments, we did consider CR-like models for SI and ZPR tasks. The performance of existing CR systems is not reliable enough in open-domain and non-English languages, because they are usually trained on a small-scale manually-annotated dataset. For instance, the advanced Stanford CoreNLP only achieve 60.0 and 53.9 F1 scores on CoNLL English and Chinese testsets, respectively. The performance is far from practical use as part of or as baselines.
>
> 1.  In SI, besides single entities, speaker mentions can also be phrases, pronouns, or multiple entities (e.g. an utterance is spoken by multiple speakers simultaneously), which can not be easily handled by CR models. CR has actually been used in previous SI rule-based methods. As there is no released code, we run our SI baseline on a public SI dataset JY[6] for fair comparisons.
> 2.  The ZPR can be considered as a particular task of CR, which resolves zero anaphora by recovering omitted pronouns. In our submission, we have re-implemented the BERT-based ZPR system (Song et al., 2020) and reported results on GuoFeng dataset.
>
> | **Task**              | **CR-based** | **GuoFeng RoBERTa (large)** |
> |-----------------------|--------------|-----------------------------|
> | SI (JY Dataset)       | 86.6         | 98.3                        |
> | ZPR (GuoFeng Dataset) | 17.4         | 34.3                        |
>
> **Q6. What are the new challenges arising from the benchmark?**
>
> We propose the GuoFeng benchmark for the target evaluation on the discourse modeling. GuoFeng is a comprehensive benchmark covering various NLP tasks, which require different discourse information to accomplish the task goal. Experimental results on the proposed benchmark show that existing pre-trained models lack the ability of modeling discourse: (1) no single model performs best in all tasks that require different types of discourse information; (2) GuoFeng models with fine-grained pretraining on document-level data consistently outperform their vanilla counterparts.
>
> Further analyses on the GuoFeng diagnostic test suit confirm our claim: existing pre-trained model can be easily attacked by simply modifying a single token that plays a discourse role. GuoFeng models can alleviate this problem, while there is still a large improvement space, indicating the necessity of more advanced techniques of modeling discourse information for pre-trained models.
>
> **[Reference]**
>
> [1] Hua He, Denilson Barbosa, and Grzegorz Kondrak. Identification of Speakers in Novels. ACL 2013.
>
> [2] Aya A Mitani, Phoebe E Freer, and Kerrie P Nelson. 2017. Summary measures of agreement and association between many raters' ordinal classifications. Annals of epidemiology.
>
> [3] Lynn Carlson and Daniel Marcu. Discourse Tagging Reference Manual.
>
> [4] Yizhong Wang, Sujian Li, Houfeng Wang. A Two-Stage Parsing Method for Text-Level Discourse Analysis. ACL 2017.
>
> [5] Stanford CoreNLP CR system: <https://stanfordnlp.github.io/CoreNLP/coref.html>.
>
> [6] Yuxiang Jia, Huayi Dou, Shuai Cao, and Hongying Zan. Speaker Identification and Its Application to Social Network Construction for Chinese Novels. IALP 2021.

---

> ### Author Response · Authors · 2022-11-18
> **Response to Reviewer eMwY (PART III)**
>
> **Q4. What are the annotation guidelines and inter-annotator agreements for the test suite?**
>
> The annotation guideline contains two types of contrastive examples: Cohesion and Coherence. For each type, we conduct a two-step process: first Tagging and then Modification.
>
> **Cohesion contrastive test examples:**
>
> 1.  **Tagging**. Given a text, linguistic experts are asked to label five general categories of cohesive devices: reference (anaphora, coreference), ellipsis (substitution), conjunction, and lexical cohesion (repetition). Here we take explicit devices in English for example, but we also label implicit ones especially in Chinese (e.g. zero anaphora).
> 2.  **Modification**. Based on the tags, we replace original cohesive devices in text with incorrect candidates to generate incorrect examples. Considering three types of tasks, we employed different strategies, varying in which part of the text to be modified (detailed in Section 3).
>
> | **Category** | **Tagging Example**                                   | **Modification Candidates**                                                                                                                                                |
> |--------------|-------------------------------------------------------|----------------------------------------------------------------------------------------------------------------------------------------------------------------------------|
> | anaphora     | {Audi}_{a0:ent} ... {It}_{a0:pro}                     | **pronoun**: I, me, you, he, she, it ... **demonstrative**: this, that, these, those ...                                                                                           |
> | coreference  | {Audi}_{cr0:ent} ... {The company}_{cr0:ent}          | noun phrase                                                                                                                                                                |
> | ellipsis     | - Will he come? - I don't know {es}_{if he will come} | **nominal**: one, ones, same ... **verbal**: do ... **clausal**: so, not ...                                                                                                           |
> | conjunction  | {Finally}_{c:temporal}, she hid in the forest.        | **additive**: and, or, thus, by the way ... **adversative**: yet, but, rather, however ... **causal**: so, then, because, therefore ... **temporal**: then, next, finally, before that ... |
> | repetition   | {dress}_{re0} ... {frock}_{re0}                       | same word/synonyms/antonyms                                                                                                                                                |
>
> **Coherence contrastive test examples:**
>
> 1.  **Tagging**. Given a text, linguistic experts are asked to parse it into a RST tree using rstWeb toolkit according to Discourse Tagging Reference Manual [3]. We mainly consider six widely-used relations [4].
> 2.  **Modification**. Based on RST trees, we change the original discourse relations to generate incorrect examples. We minimally modify and delete entity/pronoun/connective words and even the whole semantics of a span.
>
> | **Category** | **Tagging Definition (Nucleus/Satellite)**                                                       | **Modification Methods**                           |
> |--------------|--------------------------------------------------------------------------------------------------|----------------------------------------------------|
> | Attribution  | S gives the source of attribution for an instance of reported speech in N                        | replace the speaker with another one               |
> | Elaboration  | S gives further information about the content of N                                               | delete further information                         |
> | Contrast     | Two or more N contrast along some important dimension                                            | replace or add additive connective words           |
> | Reason       | S provides the reason for the action carried out in N                                            | modify the semantics of S to reduce causality      |
> | Evidence     | S provides information with the goal of convincing the R to accept the information provided in N | modify the semantics of S to reduce persuasiveness |
> | List         | A series of N is given, without contrast or explicit comparison                                  | modify one of N to break coordinating relation     |

---

> ### Author Response · Authors · 2022-11-18
> **Response to Reviewer eMwY (PART II)**
>
> **Q2. How to evaluate each proposed task?**
>
> Due to page limitation, we list automatic metrics in Table 1 and put detailed descriptions in Appendix B. We will move them to Section 2 in the revised version. Note that we only put primary metrics in Table 1 and 4, and provide results using additional metrics in Appendix B.
>
> -   **SI**: Both macro-averaged F1 and exact match (EM) are used as the evaluation metrics following standard extractive machine reading comprehension tasks (Rajpurkar et al., 2016). We used F1 as the primary metric and EM as an additional one.
> -   **ZPR**: We follow the common practice to use micro F1 score as the standard evaluation metric (Song et al., 2020). The precision and recall are used as additional metrics.
> -   **MRC**: As this is a multiple-choice test, we directly employed accuracy (acc=n/m) as an evaluation metric, where n is the correct predicted answers and m is the size of candidate set.
> -   **NT, CCT, PT**: There are a number of evaluation metrics for measuring general performance of translation systems. BLEU is the most widely-used one, which measures the precision of n-grams of the MT output compared to the reference, weighted by a brevity penalty to punish overly short translations (Papineni et al., 2002). TER is an error metric for machine translation that measures the number of edits required to change a system output into one of the references (Snover et al., 2006). METEOR incorporates semantic information by calculating either exact match, stem match, or synonymy match (Banerjee & Lavie, 2005). Furthermore, COMET is a neural framework for training multilingual MT evaluation models which obtains new SOTA levels of correlation with human judgments (Rei et al., 2020). We used BLEU as the primary metric and others as additional.
> -   **TE**: This new task can be regarded as an inverse summarization, thus we used the commonly-used BLEU metric in summarization to compare between output and reference. We also use Perplexity (PPL) as an additional metric to measure fluency.
> -   **TI, TC**: We take the following automatic metrics: PPL, BLEU, BertScore (Zhang et al., 2019a), and diversity scores (Dist) (Li et al., 2016). For Dist, we report the degree of diversity by calculating the ratio of distinct 2-grams/4-grams in the generated text. PPL is the primary metric and others are additional ones.
>
> **Q3. What is the inter-annotator agreements for the tasks requiring human annotations?**
>
> The datasets in understanding tasks require human annotations. Each instance is annotated by two linguistic experts. We followed [2] to measure the inter-annotator agreement by calculating the average pairwise Cohen’s kappa score. As seen, the annotations can reach 0.76%\~0.95% kappa scores, demonstrating that the annotators work efficiently and consistently under the annotation guideline.
>
> | **Task/Datasets** | **Agreements** |
> |-------------------|----------------|
> | SI                | 0.76           |
> | ZPR               | 0.91           |
> | MRC               | 0.97           |

---

> ### Author Response · Authors · 2022-11-18
> **Response to Reviewer eMwY (PART I)**
>
> **Q1. The detailed settings of the proposed tasks.**
>
> -   **SI: are the speakers always mentioned before the utterance of interest?** No. A speaker mention can also appear in the utterance itself, the context after the given utterance, or the utterances spoken by other speakers.
> -   **SI: what are the expected outputs if there are multiple utterances with quotation marks?** We follow previous classical SI studies (e.g. [1]) assuming that all utterances in the same paragraph are spoken by the same speaker.
> -   **ZPR: Are they Chinese movies with English subtitles, English movies with Chinese subtitles, or both?** Both. In the crawled subtitles, around 80% of the data are originally written in English and 20% of the data in Chinese. We manually checked the accuracy and consistency of discourse phenomena in sampled text for each movie and found that both types of subtitles are of high quality.
> -   **NT: Is eclipsis not important for novel translation?**  While this work mainly mentions referential and lexical cohesion, ellipsis phenomena are also significant when translating ellipsis sentences into non-ellipsis one. Taking NT dataset for instance, there are 8% Chinese/English sentences contain *nominal, verbal or clausal ellipsis.*
> -   **NT: How about concept consistency in scientice fictions?** The consistency is another critical issue in discourse-level MT, where a repeated term (e.g. concept, entity, tense) should keep the same translation throughout the whole document. This phenomenon is frequently seen in science and wuxia genres in NT dataset, leading to that MT models need to spend a substantial amount of their capacity in disambiguating and translating these terms.
> -   **CCT: do you mean traditional Chinese? Novels written in traditional Chinese?** Both traditional Chinese and simplified Chinese belong to modern Chinese, while classical Chinese is a traditional style of written Chinese that evolved from the classical language, making it different from any modern Chinese. Novels are written in modern Chinese.
> -   **PT: what's the target language?** English (as shown in “Out” in Figure 3).
> -   **How to evaluate the generation tasks if there are multiple plausible answers?** We use reference-based metrics (e.g. BLEU) to measure the quality of generated output. Indeed, for open-ended text generation that may have multiple plausible answers, reference-based metrics can give unfaithfully low scores to generated texts that are good but very different to the references. However, how to authentically evaluate the quality of open-ended text generation is challenging and remains an open question. To date, reference-based metrics are still widely used for various text generation tasks despite the limitation mentioned above. One way to improve the reliability of reference-based metrics is by collecting more and diverse references. Human evaluation may be an alternative way, however, this can be subjective and inconsistent, especially difficult for maintaining dynamic leaderboards in our case.

---

### Author Response · Authors · 2022-11-18
**General Response to All Reviewers**

We thank all the reviewers for their insightful and valuable comments which will serve to improve the paper considerably. We will attend to all comments to the best extent in the revised version.

---

### Decision · Program_Chairs · 2023-01-20

**Decision:**

Reject

**Justification For Why Not Higher Score:**

Not good enough

**Justification For Why Not Lower Score:**

NA

**Metareview: Summary, Strengths And Weaknesses:**

The benchmark, especially the dataset offered in this paper can be a good resource for document-level NLP tasks, which has not been the main focus in ML conferences. Having said that I also agree with Reviewer CWMG that it should do more to ensure the dataset quality, presentation clarity and baselines. Although some of these have been addressed in the rebuttal but the reviewers feel the paper is not yet ready to be published at its current form.

**Summary Of Ac-Reviewer Meeting:**

NA